# An Overview towards Zebrafish Larvae as a Model for Ocular Diseases

**DOI:** 10.3390/ijms24065387

**Published:** 2023-03-11

**Authors:** João Gabriel Santos Rosa, Monica Lopes-Ferreira, Carla Lima

**Affiliations:** Immunoregulation Unit of the Laboratory of Applied Toxinology (CeTICs/FAPESP), Butantan Institute, São Paulo 05503-900, Brazil

**Keywords:** zebrafish, larvae model, retina, visual impairment, diabetic retinopathy, regeneration, ocular infections

## Abstract

Despite the obvious morphological differences in the visual system, zebrafish share a similar architecture and components of the same embryonic origin as humans. The zebrafish retina has the same layered structure and cell types with similar metabolic and phototransduction support as humans, and is functional 72 h after fertilization, allowing tests of visual function to be performed. The zebrafish genomic database supports genetic mapping studies as well as gene editing, both of which are useful in the ophthalmological field. It is possible to model ocular disorders in zebrafish, as well as inherited retinal diseases or congenital or acquired malformations. Several approaches allow the evaluation of local pathological processes derived from systemic disorders, such as chemical exposure to produce retinal hypoxia or glucose exposure to produce hyperglycemia, mimicking retinopathy of prematurity or diabetic retinopathy, respectively. The pathogenesis of ocular infections, autoimmune diseases, or aging can also be assessed in zebrafish larvae, and the preserved cellular and molecular immune mechanisms can be assessed. Finally, the zebrafish model for the study of the pathologies of the visual system complements certain deficiencies in experimental models of mammals since the regeneration of the zebrafish retina is a valuable tool for the study of degenerative processes and the discovery of new drugs and therapies.

## 1. Introduction

Zebrafish (*Danio rerio*) present development of the brain, nervous system, and visual system that is similar to that of other vertebrates, which makes it a valuable model for the study of visual neuroscience. The anatomical, physiological, genetic, and behavioral components of zebrafish visual processing were studied in adults and during the embryo-larval period (Figure 1), providing an opportunity for the study of disorders related to defects in the developmental process, hereditary diseases, or provoked by environmental stressors (light, chemicals, or trauma), infections and [1,2].

Here, we revisit important insights into zebrafish visual system morphogenesis, and summarize established zebrafish models of ocular pathologies, from inherited retinal diseases to congenital or acquired malformations and ocular infections. We discuss important features of photoreceptor degenerations, as well as anterior and posterior eye diseases, along with pathophysiological mechanisms and their relation to human diseases. We also discuss the contribution of zebrafish to regenerative research and highlight the understanding of zebrafish retina regeneration, reinforcing future directions to explore ophthalmological research with zebrafish.

For this review, we searched the PubMed database and cited recent articles, most of which were published in the last 5 years. Articles previous to 2018 that are cited refer to proof-of-concept articles or consolidated concepts about anatomical features, physiological hallmarks, or pathophysiological mechanisms.

## 2. Zebrafish Ocular Development and Physiology

Eye development begins with the specification of the eye field within the anterior neural plate, which is established by the overlapping expression of several eye field transcription factors. The structures of the anterior eye include the cornea, lens, iris, ciliary body, hyaloid vasculature, and specialized structures at the iridocorneal angle, and the structures of the posterior eye include the neural retina, retinal pigment epithelium (RPE), and choroid.

The morphogenesis of the visual system in zebrafish presents similarities with humans since it is also based on the same three embryonic tissues: neuroectoderm, superficial ectoderm, and mesenchyme. The optic vesicle, a structure that derives from the forebrain, appears at approximately 12 hpf (hours post fertilization) and at 20 hpf it forms a structure composed of an internal neural ectoderm, which will give rise to the retina and external neural ectoderm, which give rise to the RPE, forming the posterior segment of the eye [3].

The retina is a structure of particular interest as it is responsible for sensory detection and damage to the retina results in loss of vision. Zebrafish retinal neurogenesis begins with the differentiation of ganglion cells at around 28 hpf in the ventronasal retina, followed by amacrine, horizontal and Müller glia cells at around 50 hpf. The exit of the axons to the optic tectum occurs around 40 hpf. Most of the lamination process of the retina occurs by 48 hpf. Between 50 and 55 hpf, the expression of opsin and rod cells begins and four different cone photoreceptor subtypes (UV-, S-, M-, and L-cones) become apparent. Synaptic structures indicative of functional maturation (ribbon triads) arises within photoreceptor synaptic terminals at around 65 hpf, followed by bipolar cell ribbon synapses at approximately 70 hpf. The signal transmission from photoreceptors to second-order neurons starts around 84 hpf and becomes fully functional at 5 dpf [4]

Structurally, the mature zebrafish retina is composed of three nuclear layers separated by the outer and inner plexiform layers (OPL and IPL, respectively). Photoreceptor cell bodies (rods and cones glutamate releasing-cells) reside in the outer nuclear layer (ONL), amacrine (local dopaminergic interneurons that provide modulation pre- and/or post-synaptically), bipolar (glutamatergic neurons that transmit light signals into the next processing layer), and horizontal (inhibitory interneurons that locally modulate photoreceptor synaptic output) occupy the inner nuclear layer (INL), and ganglion cell bodies (glutamatergic spiking neurons) are contained in the ganglion cell layer (GCL). Synapses between these nuclear layers and retinal neurons occur in the plexiform layers [1,4,5].

From GCL, ganglion cell axons emerge and form the optic nerve, which transmits information to the optic centers in the brain. Additionally, the retina contains microglia (resident immune cells located primarily in the synaptic layers) and two types of true glial cells: Müller cells (a type of radial glia) and astrocytes (associated with axons of retinal ganglion cells—RGCs) [4].

Photoreceptors (PRs) are specialized cells that compose the layered structure of the retina and promote and maintain the phototransduction process. These neuron cells comprise the inner segment (IS), where most of the organelles required for metabolic functions are found, and the outer segment (OS), which is rich in mitochondria, generating the necessary amount of energy for phototransduction. UV-cones are short single cones that express *sws1*, an opsin with maximum sensitivity in the ultraviolet range. The S-cones are long single and express *sws2*, with sensitivity to short wavelengths in blue. M-cones express *rh2* class opsins, detecting medium wavelengths in red. L-cones express an *lws*-class opsin gene, with sensitivity to longer wavelengths in green. In rod and cone photoreceptors, light triggers opsin activation followed by phototransduction and synaptic signaling [6].

The photoreceptors also closely associate with the RPE cells, which provide structural, trophic, and metabolic support and are directly involved in the recycling of opsins [6]. Damage in PRs can be caused by several cellular dysfunctions, such as non-functional ion channels and enzymes and altered phototransduction cascade components or cellular structures. Melanin in the RPE is believed to play an important role in protecting photoreceptors from light-induced damage by absorbing stray light, which would otherwise degrade retinal image quality [6]. Thus, RPE dysfunctions negatively affect the photoreceptors, ultimately leading to cell death.

The process of intraocular vascularization is initiated at 29 hpf by a population of mesodermal and neural crest cells that have migrated around the optic cup to enter the eye through the choroid fissure to form the periocular mesenchyme (POM). This mesenchyme gives rise to the first hyaloid vessels. Rapidly, the vessels organize in a hemispherical basket at the back of the lens forming the hyaloid vasculature, which is composed entirely of arterial vessels. This embryonic hyaloid vasculature is eventually replaced by the mature retinal vasculature that is tightly associated with the RGC layer [7]. The final step in the development of a functional vasculature is the formation of the blood–retinal barrier (BRB) by vascular endothelial cells lining the retinal vessels and an epithelial component formed by the RPE [8].

The zebrafish cornea is a laminated avascular structure and quite similar to that of a human, presenting the epithelium, Bowman’s layer, stroma, Descemet’s membrane, and a single layer of junctionally connected and polygonal endothelium cells. However, the zebrafish cornea has a thinner stroma and does not show corneal nerve fibers [9]. In the same way, the iridocorneal angle in zebrafish and humans is responsible for maintaining the intraocular pressure (IOP) by producing and draining aqueous humor.

Although the zebrafish lens is more spherical than a human’s and made of an outer epithelial layer, its morphological structure is remarkably similar to that of humans [7]. The lens is contained within a semi-elastic capsule and grows by depositing concentric layers of fibrous cells over existing tissue. Epithelial cells proliferate, differentiate, and elongate into lens fiber cells at the equator of the capsule. Fiber cells contain high concentrations of crystalline proteins that range from periphery to center, creating a gradient of concentration and refraction. The zebrafish lens delaminates from the overlying surface ectoderm as a solid mass of cells rather than invaginating to form a hollow vesicle [10].

Zebrafish show a high degree of binocular coordination; most of the time, the eyes are moved in a conjugate fashion with the notable exception of convergence during prey capture and spontaneous monocular saccades [11]. Recently, the oculomotor system studies were expanded by Brysch, Leyden, & Arrenberg (2019) [12], which presented evidence for a mixed mono-/binocular code in the hindbrain.

The oculomotor system for horizontal movement participates in maintaining the stability of eye positions, as well as eye movements during saccades, optokinetic, and vestibulo-ocular reflexes (OKR and VOR, respectively) and other behaviors.

Quick eye movements are important components of visual behavior, improving individual visual ability. Schoonheim, Arrenberg, Del Bene & Baier (2010) [13] described the brain region responsible for saccade movement in a larval zebrafish model, besides describing the neuron circuitry, suggesting certain homology to mammal circuits. The comprehension of complex eye movements is essential for a better understanding of ocular disorders, providing different perspectives for novel treatments.

## 3. Zebrafish as a Model for Ocular Pathologies

Zebrafish is a suitable model for the recapitulation of eye and vision diseases since it allows the sub-localization of the injury process from the visualization of the eye and its structures through the transparent larva. In this way, the underlying cellular and molecular mechanisms of pathology can be understood. Phenotype-based screening, including the assessment of the behavioral repertoire pattern of zebrafish larvae, has been applied to assess visual-neural circuits and has revolutionized the scientific environment of the pharmaceutical industry, allowing the simultaneous screening of numerous new molecules, the repositioning of already approved drugs, as well as the investigation of alternative therapies [14,15,16].

Several groups have been using mutagenesis (large-scale mutant or targeted genetic screens) or gene editing approaches to create mutants and morphants of zebrafish in order to obtain models for the study of human congenital eye diseases. More recently, the use of CRISPR-mediated gene editing technology associated with the use of morpholinos in zebrafish fluorescent reporter lines can track potentially pathogenic genetic variants as well as understand the intricate molecular signaling cascade underlying diseases provoked by environmental stressors (light, chemicals, or trauma), infections, altered metabolisms, and aging reviewed by [17,18]. With these approaches in zebrafish, congenital malformations generated from disruptions in genes important for the morphogenesis of both anterior and posterior structures of the eye during the early stages of embryonic development have been mapped.

Studies from several groups demonstrate the requirement of some genes in malformations of the anterior structures of the eye in zebrafish, recapitulating diseases such as microphthalmia (M), anophthalmia (A), and synophthlamia/cyclopia. According to Yin et al. (2014) [19], transcription factors such as retinal homeobox (Rx) are vital for the correct development of the eye, and zebrafish with mutated genes encoding these factors demonstrated that eye morphogenesis is dependent on a signaling network to regulate these factors. Besides genetic mutations, the modulation of transcription factors can be assessed with chemical exposure. Santos-Ledo et al. (2013) [20] exposed zebrafish embryos to ethanol treatment and observed that ethanol exposure disrupts the correct eye morphogenesis.

Coloboma (C) resulted from the failure of the optic fissure to close properly, and Pillai-Kastoori et al. (2014) [21] showed that the transcription factor Sox11 is required to maintain specific levels of Hedgehog signaling during ocular development; once Sox11 has mutated, zebrafish present abnormal ocular formation. Likewise, hyaloid vasculature formation is crucial for optic fissure fusion, and the action of transcription factor pax2a facilitates the remodeling process [22].

Aniridia (complete absence of the iris), corneal opacities, iris hypoplasia, aphakia (absence of the lens), and corneal dystrophies are anterior eye abnormalities, and were reproduced in zebrafish [23,24]. Shi et al. (2005) [25] described a zebrafish model of Peter’s anomaly (persistent adhesion between the cornea and lens), identifying a gene encoding a protein similar to humans, the Pitx3 protein. Microphthalmia and aphakia were modeled in *mab21l2−/−* mutant zebrafish [26]. Besides, Axenfeld–Rieger syndrome (combinations of anterior segment malformations with dental anomalies) and human congenital nystagmus (HCN)/infantile nystagmus syndrome (INS) [27,28,29] have also been recapitulated in zebrafish.

Finally, cataracts (congenital or age-related), defined as a clouding or loss of transparency in the lens, have been recapitulated in zebrafish that possess gene orthologs expressed exclusively in the lens (reviewed by [30]—Table 1).

The effect of gene knockdown or knockout on eye development has been used to investigate the crucial role of numerous genes in the development of the posterior eye that lead to several ocular dysfunctions, vision impairment, and blindness. Retinal degeneration (RD) forms a group of diseases resulting from a primary dysfunction in the RPE that are classified into three types: (i) diseases primarily affecting rods; (ii) diseases primarily affecting cones; and (iii) diseases where both photoreceptors (PRs) are affected.

Photoreceptor dystrophies are characterized by photoreceptor loss on the retina, and the disbalance of the retinal pigmented epithelium (RPE) leading to RD. In zebrafish, cones rely on RPE-expressed Cralbpa protein for chromophore regeneration and retinal function, and Schlegel et al. (2019) [42] reported that *rlbp1a* impairment disturbed the photoreceptor morphology, ultimately leading to cellular degeneration. Other factors such as growth differentiation factor 6a (gdf6a) are important to photoreceptor maintenance; Nadolski et al. (2020) [43] reported that mutated *gdf6a* in zebrafish led to photoreceptor aberrant growth and morphology.

RD encompasses several inherited retinal dystrophies such as retinitis pigmentosa (RP, an inherited heterogeneous disease), age-related macular degeneration (AMD), diabetic retinopathy (DR), achromatopsia, cone–rod dystrophy, and congenital stationary night blindness. Jia et al. (2014) [44] demonstrated that proper synaptic ribbon formation is essential for photoreceptor function. To assess this matter, the author used mutant zebrafish expressing human *Cacna1f* as a model system.

Night vision impairment is a key feature of congenital stationary night blindness (CSNB), representing a non-progressive disease caused by defective synapsis between photoreceptors and ON-bipolar cells. Bahadori et al. (2006) [45] explored the activity of nyctalopin protein on photoreceptor metabolism as a model for CSNB. Some other RD are characterized by hypopigmentation/albinism, such as Chediak–Higashi syndrome, and Hermansky–Pudlak syndrome, a disease characterized by oculocutaneous albinism, and were successfully modeled in zebrafish by Bahadori et al. (2006) [46].

Furthermore, zebrafish have greatly improved the understanding of diseases of posterior structures of the eye related to several syndromes, such as Leber’s congenital amaurosis (LCA), modeled by Minegishi, Nakaya, & Tomarev (2018) [47] through a *cct2* mutant zebrafish line, whereas Stiebel-Kalish et al. (2012) [48] studied the same disease with a *Gucy2f* zebrafish knockdown line; Stargardt’s disease, Bardet–Biedl syndrome (BBS), Usher syndrome, Joubert syndrome, Meckel-Gruber syndrome, and Senior-Loken syndrome (Table 2).

Finally, the development of methods to measure intraocular pressure (IOP) in zebrafish mutants has enhanced the offer of models for the study of the complex genetics and molecular mechanisms of congenital or age-related glaucoma. Glaucoma characterized by ganglion cell death and damage to the optic nerve can be divided into primary open-angle glaucoma, in which the iridocorneal angle is not blocked, primary closed-angle glaucoma, in which the iridocorneal angle is blocked, and secondary glaucoma, in which the glaucoma arises due to a prior insult to the eye (reviewed by [18]. Table 2).

The *World Report on Vision* [87] by WHO estimates that the leading causes of vision impairment and blindness are cataract (17.8 million and the second leading cause of moderate or severe vision impairment, with 83.2 million), glaucoma (3.6 million and the fourth leading cause of moderate or severe vision impairment, with 4.1 million), age-related macular degeneration (AMD, 1.9 million and the third leading cause of moderate or severe vision impairment, with 6.2 million), and diabetic retinopathy (DR, 1.1 million and the fifth leading cause of moderate or severe vision impairment, with 3.8 million people). The report estimates that a person can have more than one cause of vision impairment; however, epidemiological studies tend to report only the primary cause.

In this review, we will detail the research carried out in a zebrafish model that aims to elucidate the genetic and cellular mechanisms of the diseases that most affect the world population: cataract, glaucoma, age-related macular degeneration (AMD), and diabetic retinopathy (DR). Finally, we will also address the progress made in this model organism of retinitis pigmentosa (RP) and the regeneration process.

### 3.1. Cataract

Cataract is a disease characterized by visual impairment due to lens opacity. This disorder is considered the primary cause of vision loss, and is therefore a major public health issue [88]. According to disease origins, cataract can be classified as congenital, when caused by genetic mutation and consequent misfolding of the lens proteins. Physiologically, the lens is formed by laminar structures of fiber cells derived from epithelial cells. In the fiber cells that form the lens, crystallin is a key protein to maintain the optical properties and its expression represents a complete differentiated state of fiber cells [89,90].

The crystallin protein family represents about 90% of soluble proteins in the human lens, including α-, β-, and γ-crystallins [91]. According to data reviewed by Hong & Luo (2021) [15], mutated genes encoding the crystallin proteins lead to the misfolding of proteins, resulting in degraded and unfunctional proteins and, consequently, lens abnormality, resulting in visual impairment. Thus, the generation of models with mutated lens proteins represents a valuable tool to study cataract phenotypes [92,93,94].

Li et al. (2012) [31] injected mutant crystallin gamma C (crygc) mRNA into zebrafish embryos, producing a mutated crystallin protein, which resulted in a cataract phenotype. In the same way, Wu, Zou, Mishra, & Mchaourab (2018) [32] used transgenic strains of zebrafish with crystallin protein mutations, which produced lens abnormalities, similar to the findings of Mishra et al. (2018) [33], who developed a zebrafish line with the loss of function of αB-crystallin and reported that a greater part of the embryo presented lens abnormalities, suggesting that this protein is essential to lens development.

Besides crystallin, other protein types also contribute to lens formation. As reviewed by Zhang et al. (2020) [35], for the correct lens development process, an organelle-free zone is crucial in order for fiber cells to appear optically transparent. In this process, the cells’ nuclei must be removed through degradation by nuclease enzymes. In the same work, the author produced a mutant zebrafish line induced by the CRISPR-Cas9 system with silenced DNase1l1l, an essential enzyme for the lens fiber cell denucleation process. The work resulted in the successful induction of a cataract phenotype in zebrafish.

The membrane-bound organelles of fiber cells in the lens undergo degradation during terminal differentiation. Organelle degradation by PLAAT (phospholipase A/acyltransferase, also known as HRASLS)-family phospholipases is essential for achieving optimal transparency and refractive function of the lens. Morishita et al. (2021) [95] reported that Hrasls phospholipases of the Plaat1 family in zebrafish are essential for the degradation of lens organelles such as mitochondria, the endoplasmic reticulum, and lysosomes.

In the developing lens, the RNA-binding protein Rbm24 binds to a wide spectrum of lens-specific mRNAs through the RNA recognition motif and interacts with the cytoplasmic polyadenylation element-binding protein (*Cpeb1b*) and cytoplasmic poly(A)-binding protein (*Pabpc1l*) through the C-terminal region. Shao et al. (2020) [96] demonstrated in zebrafish embryos that the expression of the *rbm24* gene is restricted in differentiating lens fiber cells, and its loss of function severely compromises the accumulation of crystallin proteins in the developing lens. Moreover, Rbm24 binds to a wide spectrum of lens-specific mRNAs and interacts with key components of the cytoplasmic polyadenylation complex.

Aquaporins are proteins with water permeability regulation activities, and are, according to Vorontsova, Gehring, Hall & Schilling (2018) [34], essential to lens transparency. These authors used mutant zebrafish to assess the role of these proteins on lens development and reported that Aquaporin 0a and Aquaporin 0b knocked out zebrafish present fiber lens cell disruption and lens opacity, similar to the cataract phenotype.

During the differentiation of epithelial to fiber cells, most organelles are lost, turning into quiescent cells and are thus more vulnerable to exogen damage. Thus, cataract can also be classified as acquired, generally when derived from primary conditions. High levels of glucose (diabetic cataract), ionizing radiation (radiation cataract), and alteration in intraocular pression (glaucoma) are common triggers of cataract, besides aging (age-related cataract) (Lu et al., 2022) [97]. These primary disorders produce high levels of reactive oxygen species (ROS), which are counterbalanced by the protective antioxidant system.

GJA8 is a major gap junction protein in the vertebrate lens, and mutations in the *gja8* gene cause cataracts in humans. The well-known cataractogenesis mechanism is that mutated GJA8 leads to abnormal assembly of gap junctions, resulting in defects in intercellular communication between lens cells. In the study of Ping et al. (2021) [98], they demonstrated that the ablation of Gja8b (a homolog of mammalian *gja8*) in zebrafish led to severe defects in organelle degradation, an important cause of cataractogenesis in the developing lens.

The nuclear factor-erythroid-2-related factor 2 (Nrf2), a transcriptional factor for cell cytoprotection, is known as the master regulator of redox homeostasis. Nrf2 regulates a wide range of genes involved in cellular protection against contributing factors of oxidative stress, including aging, disease, and inflammation. Nrf2 was reported to disrupt the oxidative stress that activates nuclear factor-κB (NFκB) and pro-inflammatory mediators. One representative example is the matrix metalloproteinase 9 (MMP-9), which may decompose the extracellular matrix (ECM) of lens epithelial cells (LECs) [99].

In both primary conditions and aging, the balance between ROS production and antioxidant system failure leads to oxidative stress, causing DNA damage and lipid oxidation in fiber cells, leading to cellular protein degradation, resulting in lens opacity (reviewed by [100].

### 3.2. Glaucoma

Under physiological conditions, aqueous humor is drained in order to preserve intraocular pressure, and this drainage system is composed of a structured outflow channel consisting of the trabecular meshwork, Schlemm’s channel, and the uveoscleral channel. The elevated intraocular pressure caused by an undeveloped drainage system leads to retinal ganglion cells death, culminating with optic nerve degeneration and consequent irreversible blindness, characterizing glaucoma [101]. As reviewed by Gross & Perkins (2008) [102], zebrafish are an ideal model system to study genetic mutations and multigenic interactions that lead to glaucoma.

Skarie & Link (2009) [59] reported that a mutant zebrafish line with silenced forkhead box C1 (*foxc1*) presented altered hyaloid vasculature and arteriovenous defects, as well as ocular hemorrhages and increased vascular permeability, indicative of the disruption of basement membrane integrity, which is related to the glaucoma development process. The most striking feature of Axenfeld–Rieger Syndrome (ARS) is developmental abnormalities in the anterior segment of the eye, leading to an increased risk of early-onset glaucoma. With loss-of-function techniques in zebrafish (CRISPR and morpholino), several zebrafish models have been reported to recapitulate the phenotypes of patients with ARS attributable to the foxc1 gene mutations (reviewed by [39]. Morales-Cámara et al. (2021) [63] studied the involvement of guanylate cyclase activator 1C (*guca1c*) in the pathophysiology of primary congenital glaucoma through the generation of a guca1c knocked out zebrafish, which presented cell apoptosis and gliosis, with the Müller cells intensifying the expression of glial fibrillary acidic protein. The biochemical events led to retinal damage similar to cataract.

In zebrafish, which have all four cone subtypes, *rh2* opsin gene expression depends on homeobox transcription factors such as *sine oculis* homeobox 7 (*six*). Using a morpholino strategy, Iglesias et al. (2014) [60] produced a knockdown zebrafish with loss-of-function of (*six6b)*, which resulted in underdeveloped eyes in individuals, which is the result of an altered proliferative pattern of retinal cells, suggesting an involvement of this gene with primary open-angle glaucoma. However, the *six7* gene is found only in the ray-finned fish lineage, suggesting the existence of another evolutionarily conserved transcriptional factor(s) controlling *rh2* opsin expression in vertebrates. Ogawa et al. (2019) [103] found that the reduced *rh2* expression caused by *six7* deficiency in zebrafish by the TALEN-induced mutation method was rescued by forced expression of *six6b*, which is a *six7*-related transcription factor conserved widely among vertebrates.

Through genetic and pharmacologic modulation of yp1b1 and retinoic acid, it is possible to investigate the *CYP1B1* gene function, and Williams, Eason, Chawla, & Bohnsack (2017) [61] described that both under- and overexpression of *cyp1b1* affects the permeability of the ocular fissure, implying an ocular pressure disbalance, similar to glaucoma. To test the efficacy of neurotrophins on glaucoma treatment, Giannaccini et al. (2018) [62] exposed zebrafish larvae to an oxidative stress protocol induced by ocular injection of H_2_O_2_. The oxidative damage depleted the retinal ganglionar cells, a propitious condition to develop glaucoma.

### 3.3. Age-Related Macular Degeneration (AMD)

Cone photoreceptors are equally affected by degenerative disorders. Age-related macular degeneration (AMD) affects cone photoreceptors leading to progressive central vision loss, visual acuity impairment, and color vision defects. The degeneration impacts macular cone photoreceptors, and occurs in two forms: atrophic, also called dry AMD, where the RPE, photoreceptor and choroid go through an atrophy process, and neovascular, also called wet AMD, characterized by fluid accumulation from fragile and unfunctional new blood vessels in choroid.

RP1L1 is a component of the photoreceptor axoneme, the backbone structure of the photoreceptor’s light-sensing outer segment. Noel et al. (2020) [49] generated a *rp1l1* zebrafish mutant using CRISPR/Cas9 genome editing and found that mutant fish had progressive photoreceptor functional defects, determined by electrophysiological assessment, indicating that mutations in *rp1l1* lead to photoreceptor degenerations.

Ubiquitin protein ligase E3D (UBE3D) gene missense has been proven to be associated with neovascular AMD in the East Asian population. The ube3d gene was knocked down in zebrafish by Xia et al. (2020) [51], and increased angiogenesis was observed in ube3d morphants.

It was demonstrated by Ashikawa et al. (2017) [104] in zebrafish that the deletion of fatty acid desaturase (*fads2*), which encodes a protein that functions as both Fads1 and Fads2 in other species, enhanced apoptosis in the retina upon exposure to intense light. Similarly, pharmacological inhibition of Srebf1 enhanced apoptosis and reduced *fads2* expression in zebrafish exposed to intense light.

Mutations in human prominin 1 (*prom1*), encoding a transmembrane glycoprotein localized mainly to plasma membrane protrusions, have been reported to cause retinitis pigmentosa, macular degeneration, and cone-rod dystrophy. Lu et al. (2019) [70] utilized a zebrafish model to investigate *prom1a* and *prom1b*’s role in the retina and found that *prom1b*, rather than *prom1a*, plays an important role in zebrafish photoreceptors. The loss of *prom1b* disrupted outer-segment (OS) morphogenesis, with cones degenerating at an early age, whereas rods remained viable but with an abnormal OS, even at 9 months post fertilization.

The pathogenesis of AMD using zebrafish von Hippel-Lindau (*vhl*) mutants demonstrated the development of key aspects of the human AMD disease condition, including activation of the hypoxia-inducible factor (HIF) signaling pathway, polycythemia, excessive neovascularization, macular edema, and pronephric abnormalities [105].

The disruption of RPE is also involved in the pathophysiology of AMD, once the atrophic AMD results in photoreceptor degeneration derived from RPE loss. As demonstrated by Lu, Leach, & Gross (2022) [106] using *rpe65a*:nfsB-eGFP transgenic zebrafish and metronidazole (MTZ)-mediated cell ablation, the mTOR activation is involved in the repair response to the retinal damage initiated by Müller glia.

Rastoin, Pagès & Dufies (2020) [50] compiled data about neovascular AMD, as well as its pathophysiology and treatment. Furthermore, the authors explored the zebrafish as a model system, connecting important factors related to novel angiogenesis and the rise of AMD.

By concept, AMD is derived from photoreceptor degeneration. Thus, different assaults can trigger the degenerative process. Cheng et al. (2021) [52] exposed zebrafish to light-induced retinal damage, and reported light-induced apoptosis, oxidative stress, DNA damage and autophagy on retinal cells, suggesting the production of a zebrafish model to AMD.

### 3.4. Diabetic Retinopathy (DR)

Diabetic retinopathy (DR) associated with several diabetes-induced dysfunctions include the apoptosis of endothelial cells (ECs) and pericytes, a chronic low-grade inflammatory response with leukocyte adhesion to the retinal vasculature, increased vascular permeability, breakdown of the blood–retinal barrier (BRB), and the presence of acellular capillaries and microaneurysms. The pathogenesis of DR is multi-faceted and includes oxidative stress, pro-inflammatory changes, and advanced glycation end products (reviewed by [107]. The prevalence of DR from the year 2015–2019 has been described [108] as 27% of non-proliferative diabetic retinopathy (NPDR) covering almost up to a quarter, followed by diabetic macula oedema (DMO) at 4.6% and proliferative diabetic retinopathy (PDR) at 1.4%.

The induction of a hyperglycemic status is a valuable approach to study the pathologic alterations in zebrafish retina derived from diabetic retinopathy. Jung, Kim, Lee, & Kim (2016) [54] exposed 3 days post-fertilization (dpf) zebrafish larvae to an embryo medium with several concentrations of glucose and defined the treatment as high-glucose condition. The study demonstrated that 72 h of exposure to a concentration of 130 mM of glucose was able to provoke morphological alterations in hyaloid-retinal vessels (augmented vessel diameter), increased levels of VEGF mRNA, and irregular staining of the zonula ocludens-1 protein (ZO-1), indicating endothelial tissue disruption.

In the same way, Lee & Yang (2021) [55] exposed 3 dpf zebrafish larvae to determined concentrations of glucose for 72 h. The authors found increased retinal vessel diameter and VEGF and diminished expression of ZO-1. They also measured the retina layer thickness through histology, and despite no alterations, the TUNEL (terminal deoxynucleotidyl transferase dUTP nick end labeling) assay revealed increased apoptotic cells following glucose exposure. Hyperglycemia during embryogenesis of the retina also leads to pathological alterations in the retina vasculature.

Singh et al. (2019) [56] used different concentrations of glucose (4 and 5%) to induce hyperglycemia during embryonic development in zebrafish. The work consisted of 3 hpf embryos exposed to glucose in a fluctuating way, alternating the different concentrations of glucose every 24 h with vehicle until 120 hpf. They evidenced an alteration in the retinal vasculature, with exposure to 4% and 5% of glucose increasing the retinal blood sprouts and provoking microvascular damage, evidenced by extravasation of the hemoglobin in retinal blood vessels. Additionally, the histologic analysis showed increasing INL thickness, suggestive of Müller glial cell hypertrophy. According to Reichenbach et al. (2007), the hypertrophy and loss of metabolic function of Müller glia cells lead to progressive neuronal loss and consequent visual impairment.

Advanced glycation end products (AGEs) seem to be related to DR; the glucose metabolite methylglyoxal (MG) and its derived hydroimidazolone were found in elevated levels in the blood of patients with PDR [109]. Thus, Li, Zhao, Sang, & Leung (2019) [58] exposed 10 hpf zebrafish to different concentrations of MG until 96 hpf, in combination with glucose or not to mimic diabetes hyperglycemia. As a positive control to angiogenesis, they used GS4012 as an angiogenesis inducer. The authors reported that treatments with a higher concentration of MG (1000 µM) and MG plus glucose caused an increase in retinal blood vessel diameter and vascular density, similar to the group exposed to VEGF inducer, indicating intense angiogenesis following MG treatment.

### 3.5. Retinitis Pigmentosa (RP)

Both rods and cones PR can be affected by developmental diseases. As reviewed by Noel, Allison, MacDonald, & Hocking (2022) [53], defective cellular processes such as inefficient cell adhesion and impaired vesicle trafficking, along with ciliopathies, can drive PR diseases and degeneration, and can generally be modeled by producing mutants expressing failed visual function and ocular architecture genes.

RP is a genetically heterogeneous disease in which one of a large number of mutations causes the death of rod photoreceptors, and it is a common nomenclature for rod photoreceptor diseases and degeneration. Photoreceptor loss provokes a disruption of retina homeostasis due to an alteration of the metabolic processes, and consequently leads to retina dysfunction. Inflammatory events and the activation of glial cells (gliosis) damage the blood–retina barrier and the retinal vascular unit, causing pathological retina remodeling [110,111].

Night blindness and progressive peripheral vision impairment are signals of rod photoreceptor loss, a primary event of RP. After the complete degeneration of rod cells, visual function is maintained by cone photoreceptors, where macular cones ensure central vision. Nevertheless, secondary to RP, cone photoreceptor degeneration generally occurs, provoking total blindness disorders. Neurodegeneration caused by inherited retinal dystrophies (IRDs) leads to visual impairment and even blindness [112].

Zelinka, Sotolongo-Lopez & Fadool (2018) [64] produced a transgenic zebrafish with a specific mutation in the gene encoding the visual pigment rhodopsin (RHO), which is a common cause of retinitis pigmentosa (RP). Transgenic zebrafish expressing mutant human rhodopsin—*Tg(rho:Hsa.RH1_Q344X)*—was used by Ganzen et al. (2021) [65] as a RP model. The Q344X larvae intrinsically presented significant rod degeneration circa 120 hpf.

In the same way, Santhanam et al. (2020) [66] generated a RP model through a transgenic line of zebrafish expressing a mutation that results in a misfolded rhodopsin, which activates the unfolded protein response followed by proteasomal degradation, leading to rod cell death. Furthermore, it has been demonstrated that the death of rods reduces oxygen consumption, resulting in residual excess oxygen in the outer retina and the consequent production of superoxide radicals. High levels of superoxide radicals overload the antioxidant defense system and generate more reactive species, including peroxynitrite. This results in progressive oxidative damage to the cones, which contributes to their death and loss of function (reviewed by [67].

Cyclic nucleotide phosphodiesterase (PDEs) is an enzyme family that catalyzes nucleotide hydrolysis, and PDE6a plays a key role in the phototransduction process. In humans, mutated *pde6a* leads to RP. The crucial role of PDE6a was demonstrated in a zebrafish transgenic strain with a mutated PDE6a protein in retinal rods, which presented degeneration of rods and consequent vision impairment [68].

Kawase et al. (2016) [69] used light-induced retinal damage (LIRD) in 72 hpf zebrafish larvae, aiming to compare and identify deregulated genes in several models of LIRD. To perform the test, zebrafish larvae were held in the absence of light from 72 to 120 hpf, and then exposed to intense light (13,000 lux) for 24 h. The zebrafish larvae remained in the presence or absence of the histone acetyltransferase (HAT) EP300 inhibitor C646. The treatment was able to produce increasing apoptosis, photoreceptor degeneration, and Müller cell proliferation. They found that EP300 protects the retina against light-induced damage, acting as an anti-apoptotic agent.

In order to evaluate whether the cellular retinaldehyde-binding protein (CRALBP) is involved in pigment regeneration in the visual system, Schlegel, Ramkumar, von Lintig, & Neuhauss (2021) [42] produced a knockout (KO) zebrafish line for *rlbp1a* and *rlbp1b*, genes that encode for CRALBP in retinal pigmented epithelium (RPE) cells and Müller glia cells (MGCs), respectively. To assess the effect of CRALBP on retinoid metabolism, the larvae were exposed to a LIRD protocol consisting of different light exposures (30 min or 1 h exposure to 20,000 lux, with adaptation to the dark or not). They found that in *rlbp1a*-KO, accumulation of subretinal lipid deposits and dimorphic outer segments (OS) occurred, indicating the abnormal development or death of photoreceptors in 120 hpf larvae. Thus, the protocol of LIRD was able to produce retinal damage similar to that found in genetic disorders such as retinitis pigmentosa.

Thus, Zhang et al. (2021) [71] executed a large-scale drug screening with a zebrafish larvae model of RP using a transgenic line (rho:YFP-NTR) that express a yellow fluorescent protein (YFP) and a bacterial nitroreductase (NTR) enzyme in rod photoreceptors. This inducible model of RP was effective to implicate Poly (ADP-ribose) polymerase (PARP) as a key mediator of NTR/Mtz-mediated rod cell ablation.

## 4. Ocular Infections

While chronic eye diseases such as cataract, glaucoma, age-related macular degeneration, and diabetic retinopathy now account for a greater proportion of blindness and vision impairment globally and have their cellular and molecular mechanisms investigated in at least one zebrafish model, other treatable conditions equally important in inducing vision disturbances and blindness are still waiting to be recapitulated in the zebrafish model. Among the common eye conditions causing vision impairment, corneal injury can increase the risk of infections and keratitis.

Some of the most common pathogens that are associated with bacterial keratitis particularly when the corneal epithelial barrier is compromised or injured include Gram-positive organisms, such as coagulase-negative Staphylococcus, (e.g., Staphylococcus epidermidis), and the common Gram-negative strain Pseudomonas, *Chlamydia trachomatis*, *Onchocerca volvulus* (transmitted by the blackfly *Simulium damnosum*), virus (herpes simplex), filamentous fungi such as *Fusarium* spp. and *Aspergillus* spp., protozoa such as acanthamoeba, tuberculosis, and *Toxoplasma gondii*.

However, it is worth mentioning that infectious causes such as trachoma and onchocerciasis have declined in the last 30 years in the world [113] while Zika virus (ZIKV), a positive-sense, single-stranded, enveloped RNA virus belonging to the *Flaviviridae* family, has provided a large number of cases of congenital Zika syndrome due to vertical transmission characterized by severe microcephaly and ocular abnormalities, including chorioretinal atrophy with a hyperpigmented border and focal pigment at the macula, optic nerve abnormalities, and lens dislocation [114].

Recently, we recapitulated for the first time ZIKV infection using zebrafish embryos and larvae, especially the ability of ZIKV to infect head and eye structures during embryonic development, leading to developmental changes characterized by a reduction in head size and increased thickness of the retinal cell layers with consequent impairment of neurological and visual functions [115]. *Toxoplasma gondii* tachyzoites were demonstrated to infect cardiac myocytes, liver, spleen, brain, ovaries, pancreas, kidney and skeletal muscles, gills, and eyes of adult zebrafish; clinical signs presented in these fish included bilateral exopthalmia [116].

Takaki, Ramakrishnan, & Basu (2018) [117] developed a model for ocular tuberculosis by artificial infection with *Mycobacterium marinum (M. marinun)*, by injecting the pathogen into the caudal vein of 48 and 72 hpf zebrafish larvae. The *M. marinum* is a classical pathogen to study tuberculosis in zebrafish, leading to ocular granulomas in zebrafish similar to that in humans. Additionally, they verified that the pathogen is able to infect the eyes even with an intact blood–retina barrier (BRB). Li & Hu (2012) [118] studied the pathophysiology of *Staphylococcus aureus (S. aureus*) infection in a multi-site infection approach, inoculating the pathogen in distinguished locations of 36 hpf zebrafish. The microinjection of *S. aureus* in the eye led to the local growth of bacteria with intense migration of neutrophils followed by macrophages, and the eye was also affected by yolk circulation valley injection.

Although *Chlamydia* [119], *Onchocerca volvulus* [120], and HSV-1 virus [121] research were conducted in the zebrafish model and recapitulated the infection and the concomitant immune and anti-viral responses, ocular abnormalities were not mimicked in this model.

Detailed analyzes have shown that many of the developmental features of eye formation are conserved between zebrafish and humans. However, there are notable differences in both the development and the ability of infectious agents to infect zebrafish. But still research has shown that embryos, larvae, or adult zebrafish are useful models for studying eye diseases that cause blindness or impairment of vision.

## 5. Zebrafish Larvae as a Model for Retina Regeneration

The zebrafish ability to regenerate the retina after an injury represents an advantage in studies of retina degeneration associated with inherited retinal dystrophies (IRDs) and traumatic optic neuropathy, as well as glaucoma and diabetic retinopathy, enabling the application of zebrafish larvae on drug-screenings for IRD treatments (reviewed by [122]).

The mature adult zebrafish retina never stops developing and has a remarkable regenerative capacity. All types of retinal cells are constantly generated in the circumferential germinal zone or ciliary marginal zone (CMZ), and rod photoreceptors originating from rod precursor cells of the inner retina are added throughout life. In addition to the essential role in the maintenance of ionic homeostasis, metabolic support of neurons, uptake and recycling of neurotransmitters from synapses, Müller glia cells of the inner nuclear layer are capable of asymmetric division to generate committed progenitors of all types of retinal cells in response to injury [123].

Pioneering work demonstrated that Müller glia expressing α1T, a neuron-specific microtubule protein induced in the developing and regenerating central nervous system (CNS), can de-differentiate and become multipotent in the injured zebrafish retina [124]. Then, Bernardos et al. (2007) found that Müller glia also express the multipotent progenitor marker Pax6 (paired box gene 6), and Müller glia-derived progenitors expressing Crx (cone-rod homeobox) have a low level of proliferation in the uninjured retina, and remain competent to regenerate missing retinal neurons. Photoreceptor degeneration and other retinal cellular damage provoke a disruption in Notch signaling and cell communication, producing a series of chemical cues that interact with Müller glia, and signalize to alter several gene expression patterns and activate the cell, in a process known as gliosis.

Asymmetric, self-renewal divisions of injury-induced Müller glia and N-cadherin-mediated adhesion are required in the regeneration of neurons, as found by Nagashima, Barthel, & Raymond (2013) [125]. Dying neurons release signaling factors such as tumor necrosis factor alpha (TNFα), that up-regulate the Müller glia reprogramming genes *ascl1a* and *stat3* [126]. Likewise, TNFα can also activate microglia, and Munzel, Becker, Becker, & Williams (2014) [127] demonstrated that inflammatory cytokines produced by activated microglia, such as interleukin 6 (IL-6) and IL-11, are related to Müller glia reprogramming.

The pancreatic transcription factor mutant *pdx1^−/−^* zebrafish, described to show a diabetic phenotype, exhibits functional deficits including a significant reduction in the b-wave amplitude accompanied by retinal vascular abnormalities and the degeneration of rods and cones [128].

Molecular pathways such as ciliary neurotrophic factor [129], TGF- [130], and fibroblast growth factor (Fgf) [131], as well as the β-catenin/Wnt signaling [132] and the transcription factor Ascl1a [133], are associated with the regeneration of the zebrafish retina.

Using light-induced photoreceptor damage in larval zebrafish, Craig et al. (2010) [134] showed that light-induced photoreceptor death increased secreted β-galactoside binding protein, Galectin 1-like 2 (Drgal1-L2) from microglia and proliferating Müller glia, as well as mitotic progeny. Drgal1-L2 was demonstrated to be involved in rod regeneration, since morpholino knockdown of Drgal1-L2 resulted in reduced regeneration of rods, but not cones.

## 6. Conclusions

The unique features of the teleost eye include continued retinal growth and plasticity during post-embryonic changes and development. Zebrafish represent a useful model for recapitulation of hereditary genetic ocular abnormalities and diseases of posterior and anterior structures that significantly compromise vision and negatively affect the quality of life.

Despite all the research on neurodegenerative diseases and regeneration using the zebrafish as a model, a complete picture of all molecular factors involved in the pathophysiology of ocular diseases has not yet been achieved. However, utilizing recent advances in genome sequencing that have provided access to the complete zebrafish genome and the ability to edit genes of interest have served to expand our ability to understand, at a mechanistic level, the triggering factors as well as their interrelation with other associated endocrine factors such as metabolic disorders (obesity and diabetes).

Interestingly, different aspects of myopia, the most common human eye disease worldwide with increasing incidence, and ocular surface diseases have been established using the zebrafish model, bugeye/lrp2 mutants [135]. It was demonstrated that *lrpap1* deficiency could lead to myopia through TGF-β-induced apoptosis signaling [136].

Nevertheless, eye diseases including dry eye disease and pterygium, as well as injuries and trauma, which can even result in blindness, remain a major challenge in ophthalmological studies, including modeling in zebrafish.

It is expected that, based on experimental experiences with zebrafish, the results achieved will allow future translational research and indicate the direction we should follow to overcome the limitations of using this model organism.

## Figures and Tables

**Figure 1 ijms-24-05387-f001:**
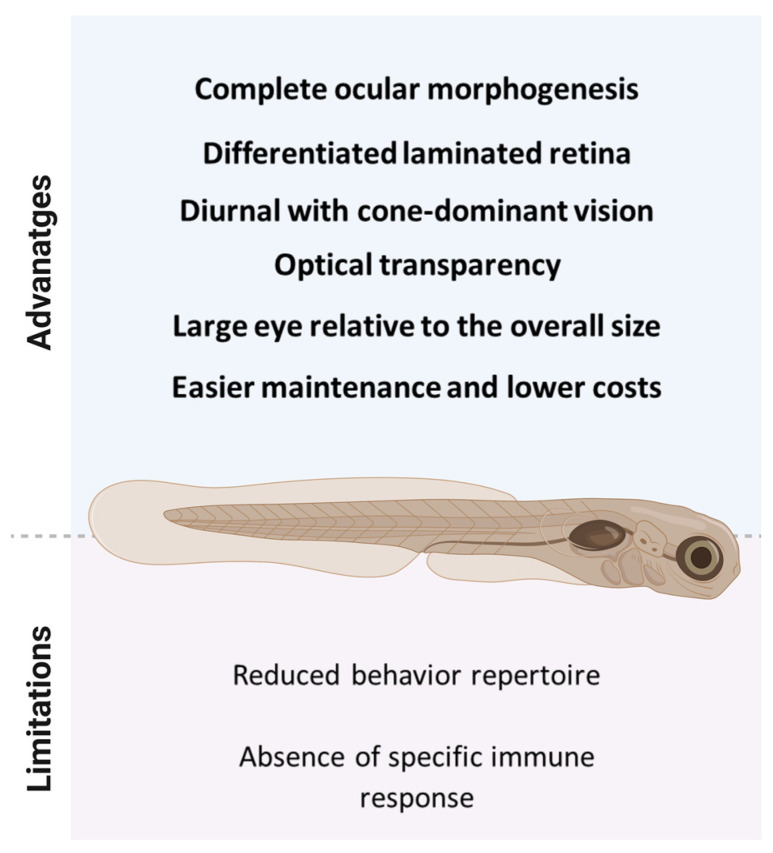
Experimental advantages and limitations of 72 h post-fertilization zebrafish larvae.

**Table 1 ijms-24-05387-t001:** Defects of the Anterior Eye.

Ocular Disease	Reference	Model	Zebrafish Features *	Human Features *
Cataracts (congenital or age-related)	Morris, 2011 [30]	Extensive review		
	Li et al., 2012 [31]	Mutation of *crygc*	Embrionic lens defects	Congenital cataract phenotype
	Wu, Zou, Mishr & Mchaourab, 2018 [32]	Mutation of *cryga* and *crygb*		
	Mishra et al., 2018 [33]	Mutation of *crygb*		
	Vorontsova, Gehring, Hall, & Schilling, 2018 [34]	Mutation of *aqp0a* and *aqp0b*		
	Zhang et al., 2020 [35]	*dnase1l1l* knockout	Lens denucleation defect and cataract	Not described
Anophthalmia (A)	Yin et al., 2014 [19]	*Mutation of rx3*	Anophthalmia and expanded forebrain	Microphthalmia and anophthalmia
Synophthlamia/cyclopia	Santos-Ledo et al., 2013 [20]	*six3a, rx3* and *rx1* disruption	Cyclopia	Holoprosencephaly and cyclopia
	Swartz et al., 2013 [36]	*vangl2*, *plk1*, *hinfp*, *mars*, and *foxi1* disruption	Synophthalmia and narrowing of the palatal skeleton	Not described
Coloboma (C)	Pillai-Kastoori et al., 2014 [21]	Mutation of *sox11*	Delayed and abnormal lens formation, coloboma, and reduction in rod photoreceptors	Coloboma phenotype
	Weaver, Piedade, Meshram, & Famulski, 2018 [22]	*pax2a* depletion	Optic fissure (OP) failure	Not described
	Ouyang et al., 2022 [37]	Loss-of-funtion of *hnrnph1*	Coloboma	High myopia
Corneal opacities	Reis et al., 2019 [38]	Mutation of *wdr37*	Cataract, microphthalmia, glaucoma, corneal clouding	Corneal opacity, coloboma, microcornea
Iris hypoplasia	Chawla, Swain, Williams, & Bohnsack, 2018 [23]	Increase/decrease in retinoic acid modulating *myoc* and *pitx2*	Changes in the ventral iridocorneal angle and decreased aqueous outflow	Corneal, iris, and trabecular meshwork abnormalities
Aniridia	Seese et al., 2021 [24]	Loss-of-function of *mab21l1*	Congenital glaucoma aphakia, malformed retina, and abnormally thick cornea, severe microphthalmia, disorganized retinal lamination, abnormal anterior structures	Microphthalmia, variable aniridia, coloboma, microcornea, lens defects (microspherophakia, cataracts) and nystagmus
Peter’s anomaly	Shi et al., 2005 [25]	Mutation of *pitx3*	Lens or other anterior segment defects	Anterior segment dysgenesis, Peter’s anomaly, and cataracts.
Axenfeld–Rieger syndrome	Reviewed by French, 2021 [39]	Extensive review		
Microphthalmia (M)	Casey et al., 2011 [40]	Mutation of *stra6*	Congenital eye malformations	Non-syndromic anophthalmia
Aphakia	Gath & Gross, 2019 [26]	Loss-of-function of *mab21l2*	Lens and retina defects, coloboma	Defects of lens development
Corneal dystrophies	Oliver et al., 2015 [41]	Mutation of *col17a1a*	Not described	Epithelial recurrent erosion dystrophy (ERED)
Human congenital nystagmus (HCN)/infantile nystagmus syndrome (INS)	Huang et al., 2006 [27]	Defective retinotectal projections	Strong spontaneous eye oscillations	Congenital nystagmus
	Maurer, Huang, & Neuhauss, 2011 [28]			
	Huber-Reggi et al., 2011 [29]			

* Features specifically related to eye disorders.

**Table 2 ijms-24-05387-t002:** Defects of the Posterior Eye.

Ocular Disease	Reference	Model	Zebrafish Features *	Human Features *
Age-related macular degeneration (AMD)	Noel et al., 2020 [49]	Mutation of rp1l1	Photoreceptor degeneration	Photoreceptor diseases
	Rastoin, Pagès, & Dufies, 2020 [50]	Extensive review		
	Xia et al., 2020 [51]	Knockdown of *ube3d*	Neovascular AMD	Oxidative damage in retinal pigment epithelium (hRPE) (in vitro)
	Cheng et al., 2021 [52]	Blue light-induced retinal damage	Retinal degeneration	Not described
	Noel, Allison, MacDonald, & Hocking, 2022 [53]	Extensive review		
Diabetic retinopathy	Jung, Kim, Lee, & Kim, 2016 [54]	Glucose exposure	Neovascularization, dilation of hyaloid-retinal vessels	Diabetic retinopathy
	Lee & Yang, 2021 [55]			
	Singh et al., 2019 [56]		Neovascularization, dilation of hyaloid-retinal vessels, and microvascular alterations	
	Reichenbach et al., 2007 [57]	Extensive review		
	Li et al., 2019 [58]	Methylglyoxal exposure	Neovascularization, dilation of hyaloid-retinal vessels	Diabetic retinopathy
Glaucoma	Skarie & Link, 2009 [59]	Knockdown of *foxc1*	Disruption of vascular endothelial tissue	Glaucoma
	Iglesias et al., 2014 [60]	Knockdown of *six6b*	Small eye phenotype	Not described
	Williams, Eason, Chawla, & Bohnsack, 2017 [61]	Manipulation of *cyp1b1*	Regulation of ocular fissure closure	Mutation on cyp1b1 is related to primary infantile-onset glaucoma
	Giannaccini et al., 2018 [62]	Exposure to H_2_O_2_	Oxidative stress-induced damage	Not described
	Cavodeassi & Wilson, 2019 [18]	Extensive review		
	Morales-Cámara et al., 2020 [63]	Knockout of *guca1c*	Retinal ganglion cell apoptosis	Not described
Retinitis pigmentosa (RP)	Zelinka, Sotolongo-Lopez, & Fadool, 2018 [64]	Mutations of *rh1–1*	Rod degeneration	Autosomal dominant retinitis pigmentosa
	Ganzen et al., 2021 [65]	*rho*:NTR zebrafish line	Rod ablation	Not described
	Santhanam et al., 2020 [66]	Mutation of *p23h*	Rod degeneration	Not described
	Campochiaro & Mir, 2018 [67]	Extensive review		
	Crouzier et al., 2021 [68]	Mutation of *pde6a*	Photoreceptor degeneration	Retinitis pigmentosa
	Kawase et al., 2016 [69]	Ligh-induced retinopathy		Not described
	Lu et al., 2019 [70]	Mutation of *prom1*		Retinitis pigmentosa, macular degeneration, and cone–rod dystrophy
	Noel et al., 2020 [49]	Mutation of *rp1l1*		Photorecepetor disease
	Schlegel, Ramkumar, von Lintig, & Neuhauss, 2021 [42]	Mutation of rlbp1		
	Zhang et al., 2021 [71]	*rho*:NTR zebrafish line	Rod ablation	Not described
Achromatopsia	Kennedy et al., 2007 [72]	No optokinetic response of mutant (*nof*)	Achromatopic blindness	Not described
	Stearns, Evangelista, Fadool, & Brockerhoff, 2007 [73]	Mutation of *pde6c*	Blindness (degeneration of cone photoreceptors)	Achromatopsia
	Viringipurampeer et al., 2014 [74]			
	Huang et al., 2018 [75]	Mutation of *per2*	Reduced vision behavior	Not described
Cone–rod dystrophy	Iribarne et al., 2017 [76]	Zebrafish *Gold Rush* mutant (*aipl1* mutant)	Cone-specific degeneration	Leber congenital amaurosis 4 (LCA4)
	Daly et al., 2017 [77]	Zebrafish *dying on edge (dye)* mutant	Defective visual behavior, altered retina morphology, photoreceptor degeneration	Visual impairment
	Schlegel et al., 2019 [78]	Knockout of *cacna2d4b*	Electroretinogram impaired response (defective phototransduction)	Retinal dysfunction (impaired cone vision)
	Nadolski et al., 2020 [43]	Mutation of *gdf6a*	Microphthalmia, photoreceptor degeneration	Not described
Congenital stationary night blindness	Bahadori et al., 2006 [46]	Zebrafish *fade out (fad)* mutant	Retina structural defects, photoreceptor degeneration	Hermansky–Pudlak syndrome (HPS)
	Jia et al., 2014 [44]	Zebrafish *wait until dark (wud)* mutant	Electroretinogram impaired response (defective phototransduction)	Congenital stationary night blindness type 2 (CSNB2)
Leber’s congenital amaurosis (LCA)	Stiebel-Kalish et al., 2012 [48]	Knockdown of *gucy2df*	Shortening of cone and rod outer segments.	Leber congenital amaurosis-1 (LCA1).
	Minegishi, Nakaya, & Tomarev, 2018 [47]	Mutation of *cct2*	Microphthalmia	
Bardet–Biedl syndrome (BBS)	Castro-Sánchez et al., 2019 [79]	Mutation of *bbs*	Microphthalmia	Ciliopathy
	Song et al., 2020 [80]		Photoreceptor degeneration	
Usher syndrome	Gopal et al., 2015 [81]	Knockout of *clrn1*	Ciliopathy	Ciliopathy
	Miles, Blair, Emili, & Tropepe, 2021 [82]	Mutation of *pcdh15b*	Loss of photoreceptor integrity	Blindness associated with Usher syndrome type 1 (USH1)
Joubert syndrome	Song & Perkins, 2018 [83]	Mutation of *arl13a*	Slow progressive photoreceptor degeneration	Joubert syndrome phenotype
	Liu, Cao, Yu, & Hu, 2020 [84]	Knockout of *tmem216*	Photoreceptor degeneration	
	Rusterholz, Hofmann, & Bachmann-Gagescu, 2022 [85]	Extensive review		
Meckel-Gruber syndrome	Lessieur et al., 2019 [86]	Zebrafish *cep290^fh297/fh297^* mutant	Slow progressive photoreceptor degeneration	Meckel-Gruber syndrome phenotype
Hermansky-Pudlak syndrome	Bahadori et al., 2006 [46]	Zebrafish *fade out (fad)* mutant	Retina structural defects, photoreceptor degeneration	Hermansky–Pudlak syndrome (HPS)

* Features specifically related to eye disorders.

## Data Availability

Not applicable.

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
