# Peer review of "An Overview towards Zebrafish Larvae as a Model for Ocular Diseases"

_ijms, 2023, doi:10.3390/ijms24065387_

Round 1
Reviewer 1 Report
The authors present their review about zebrafish larvae as a model for ocular diseases.
This is a very well written manuscript with ample references, clear tables and a correct interpretation of the literature. Therefore, I believe this manuscript can be accepted after making some important modifications (see below).
Some suggestions:
- methods: search strategy to find relevant studies? PRISMA guidelines? please add the search strategy. => https://www.prisma-statement.org/?AspxAutoDetectCookieSupport=1
- limitation: there has already been a nice paper on this subject https://www.mdpi.com/1424-8247/14/8/716 => please indicate what your study adds to the field?
- important: when you are talking about zebrafish larvae, it is important to mention the adult models of eye disease (briefly) as well. What are the advantages and disadvantages of larvae vs adult (in studying eye diseases)? I would suggest that you make a graphical/figure about this.
- table 1 and table 2: you mention the diseases + referenes => it would however be insightful to mention the major feature(s) in zebrafish larvae and how this mimicks the human phenotype; e.g. something like https://www.mdpi.com/1422-0067/22/24/13356 figure 1 and 2.
- minor comments:
o please mention a relevant model of saccades: “The oculomotor system for horizontal movement participates in maintaining the sta-122 bility of eye positions, as well as eye movements during saccades, optokinetic and ves-123 tibulo-ocular reflexes (OKR, VOR, respectively)” => Schoonheim PJ, Arrenberg AB, Del Bene F, Baier H. Optogenetic localization and genetic perturbation of saccade-generating neurons in zebrafish. J Neurosci. 2010 May 19;30(20):7111-20. doi: 10.1523/JNEUROSCI.5193-09.2010. PMID: 20484654; PMCID: PMC3842466.
o I think that “missing” some important references can be avoided by using the PRISMA guidelines for search strategy (see first comment)
Author Response
Reviewer #1
Comments and Suggestions for Authors
The authors present their review about zebrafish larvae as a model for ocular diseases.
This is a very well written manuscript with ample references, clear tables and a correct interpretation of the literature. Therefore, I believe this manuscript can be accepted after making some important modifications (see below).
Answer: the authors appreciate your willingness to review the paper and your valuable suggestions.
Some suggestions:
- methods: search strategy to find relevant studies? PRISMA guidelines? please add the search strategy. => https://www.prisma-statement.org/?AspxAutoDetectCookieSupport=1
A: Our review is a narrative review of literature encompassing the current knowledge on zebrafish larvae as a model to ophthalmological research. The aim is informing the state-of-the-art of the subject to serve as theoretical foundation for future studies. The manuscript does not mean to evaluate results of research methodologies or techniques (Grant & Booth, 2009). We selected high-quality and recent articles on the topic, to help optimize the understanding of the matter. To assure the relevance of this review, we added the following paragraph on the manuscript, at line 49:
“For this review, we searched the PubMed database and cited recent articles, most of which were published in the last 5 years. Articles previous to 2018 that are cited refer to proof-of-concept articles or consolidated concepts about anatomical features, physiological hallmarks or pathophysiological mechanisms.”
Grant MJ, Booth A. A typology of reviews: an analysis of 14 review types and associated methodologies. Health Info Libr J. 2009 Jun;26(2):91-108. doi: 10.1111/j.1471-1842.2009.00848.x. PMID: 19490148.
- limitation: there has already been a nice paper on this subject https://www.mdpi.com/1424-8247/14/8/716 => please indicate what your study adds to the field?
A: In this review, we focused on a general overview about diseases in both anterior and posterior chamber of the eye, from inherited retinal diseases to congenital or acquired malformations and ocular infections. We meant to cover the main zebrafish models for eye diseases available, besides point the undeniable advantages of zebrafish larvae to ophthalmological studies. Moreover, in the cited paper, the authors applied the knowledge of zebrafish visual system and disease models to a pharmaceutical approach, emphasizing the use of the model to drug screenings, discover of new molecules and reposition of already approved drugs.
- important: when you are talking about zebrafish larvae, it is important to mention the adult models of eye disease (briefly) as well. What are the advantages and disadvantages of larvae vs adult (in studying eye diseases)? I would suggest that you make a graphical/figure about this.
A: A figure was inserted to represent experimental advantages and limitations of zebrafish larvae.
- table 1 and table 2: you mention the diseases + referenes => it would however be insightful to mention the major feature(s) in zebrafish larvae and how this mimicks the human phenotype; e.g. something like https://www.mdpi.com/1422-0067/22/24/13356 figure 1 and 2.
A: Table 1 and table 2 were adjusted to comprise major features of each mentioned ocular disease, both in zebrafish and human.
- minor comments:
o please mention a relevant model of saccades: “The oculomotor system for horizontal movement participates in maintaining the sta-122 bility of eye positions, as well as eye movements during saccades, optokinetic and ves-123 tibulo-ocular reflexes (OKR, VOR, respectively)” => Schoonheim PJ, Arrenberg AB, Del Bene F, Baier H. Optogenetic localization and genetic perturbation of saccade-generating neurons in zebrafish. J Neurosci. 2010 May 19;30(20):7111-20. doi: 10.1523/JNEUROSCI.5193-09.2010. PMID: 20484654; PMCID: PMC3842466.
A: The authors appreciate the valuable suggestion. The follow paragraph was inserted on manuscript, at line 148:
“Quick eye movements are important components of visual behavior, improving the individuals visual ability. Schoonheim, Arrenberg, Del Bene & Baier (2010) described the brain region responsible for saccades movement in a larval zebrafish model, besides describing the neuron circuitry, suggesting certain homology to mammals circuits. The comprehension of complexes eye movements is essential to better understanding ocular disorders, providing different perspectives for novel treatments.”
o I think that “missing” some important references can be avoided by using the PRISMA guidelines for search strategy (see first comment)
A: Please see first comment.

Reviewer 2 Report
The manuscript is an excellent review on zebrafish washing as a model for studying ocular diseases. The review is comprehensive, very well structured, well written, and easy to read.
There are some minor points that must be edited to adapt to the Journal: for example, section 2.6 that should not be capitalized, and references 24 and 135 that should be edited; and references 101 and 102 which are mixed. Please pay attention to the control of all DOIs.
Author Response
Reviewer 2
Comments and Suggestions for Authors
The manuscript is an excellent review on zebrafish washing as a model for studying ocular diseases. The review is comprehensive, very well structured, well written, and easy to read.
There are some minor points that must be edited to adapt to the Journal: for example, section 2.6 that should not be capitalized, and references 24 and 135 that should be edited; and references 101 and 102 which are mixed. Please pay attention to the control of all DOIs.
Answer: the authors appreciate your willingness to review the paper and your valuable recommendations.
Section 2.6 was corrected and altered to the current paper format.
References were corrected.

Reviewer 3 Report
The authors in the manuscript entitled “An overview towards zebrafish larvae as a model to ocular diseases” have explained about the zebrafish model for the study of pathologies of the visual system complements certain deficiencies in experimental models of mammals, since the regeneration of the zebrafish retina becomes a valuable tool for the study of degenerative processes and the discovery of new drugs and therapies.
The paper is written concisely and briefly and provide significant detail on the topic which is scientifically sound. The manuscript has been written well and the content is comprehensive.
There are some issues with this article, if these issues are going to resolve then the quality of the paper is suitable for publication.
1) In a part of the introduction, should be crisp and brief about the focused study.
2) Recent references should be included.
3) There are a few typos and English and grammar errors that should be rectified.
To conclude, this is a well-written and comprehensive review article that makes a useful contribution to the field of disease model research. The quality of the review is suitable for publication in the present form after minor revision.
Author Response
Reviewer 3
Comments and Suggestions for Authors
The authors in the manuscript entitled “An overview towards zebrafish larvae as a model to ocular diseases” have explained about the zebrafish model for the study of pathologies of the visual system complements certain deficiencies in experimental models of mammals, since the regeneration of the zebrafish retina becomes a valuable tool for the study of degenerative processes and the discovery of new drugs and therapies.
The paper is written concisely and briefly and provide significant detail on the topic which is scientifically sound. The manuscript has been written well and the content is comprehensive.
Answer: the authors appreciate your willingness to review the paper and your valuable recommendations.
There are some issues with this article, if these issues are going to resolve then the quality of the paper is suitable for publication.
1) In a part of the introduction, should be crisp and brief about the focused study.
A: the follow paragraph was inserted on the manuscript, at line 37, in the new section called ‘Introduction’:
“In this review, we revisit important insights of zebrafish visual system morphogenesis, and summarize established zebrafish models of ocular pathologies, from inherited retinal diseases to congenital or acquired malformations and ocular infections. We discuss important features of photoreceptor degenerations, as well as anterior and posterior eye diseases, along with pathophysiology mechanisms and relation to human diseases. We also discuss the contribution of zebrafish to regenerative research and highlight the understanding of zebrafish retina regeneration, reinforcing future directions to explore the ophthalmological research with zebrafish.”
2) Recent references should be included.
A: 53% of references are from the last 5 years, and the follow paragraph was added to enlightenment (line 49):
“For this review, we searched the PubMed database and cited recent articles, most of which were published in the last 5 years. Articles previous to 2018 that are cited refer to proof-of-concept articles or consolidated concepts about anatomical features, physiological hallmarks or pathophysiological mechanisms.”
3) There are a few typos and English and grammar errors that should be rectified.
A: the manuscript was reviewed by the authors and the English and grammar errors were corrected.
To conclude, this is a well-written and comprehensive review article that makes a useful contribution to the field of disease model research. The quality of the review is suitable for publication in the present form after minor revision.

Reviewer 4 Report
The proposed review paper is of interest and well written.
Author Response
Reviewer 4
Comments and Suggestions for Authors
The proposed review paper is of interest and well written.
A: the authors appreciate your willingness to review the paper.
